# BEEM: Boosting Performance of Early Exit DNNs Using Multi-Exit Classifiers as Experts

**Divya Jyoti Bajpai & Manjesh Kumar Hanawal**
Department of Industrial Engineering and Operations Research
Indian Institute of Technology Bombay
Powai, Maharashtra, India
{divyajyoti.bajpai, mhanawal}@iitb.ac.in

## Abstract

Early Exit (EE) techniques have emerged as a means to reduce inference latency in Deep Neural Networks (DNNs). The latency improvement and accuracy in these techniques crucially depend on the criteria used to make exit decisions. We propose a new decision criterion BEEM where exit classifiers are treated as experts and aggregate their confidence scores. The confidence scores are aggregated only if neighbouring experts are consistent in prediction as the samples pass through them, thus capturing their ensemble effect. A sample exits when the aggregated confidence value exceeds a threshold. The threshold is set using the error rates of the intermediate exits aiming to surpass the performance of conventional DNN inference. Experimental results on the COCO dataset for Image captioning and GLUE datasets for various language tasks demonstrate that our method enhances the performance of state-of-the-art EE methods, achieving improvements in speed-up by a factor $1.5\times$ to $2.1\times$. When compared to the final layer, its accuracy is comparable in harder Image Captioning and improves in the easier language tasks. The source code for this work is publicly available at https://github.com/Div290/BEEM1/tree/main.

## 1 Introduction

Transformer-based models (Devlin et al., 2018; Radford et al., 2019; Cornia et al., 2020; Luo et al., 2021; Li et al., 2022; 2023) have set new benchmarks in performance across diverse tasks and domains through their prowess in capturing semantic information and dependencies using attention mechanisms (Vaswani et al., 2017). However, the scale and complex structure of these models poses a challenge, particularly in terms of inference speed, limiting their practicalities in resource-constrained scenarios. Also, these models are susceptible to overthinking issues (Zhou et al., 2020; Zhu, 2021) which degrades their performance in terms of accuracy and inference speed.

To address these challenges, various techniques have been proposed, including direct network pruning (Zhu & Gupta, 2017; Fan et al., 2019; Michel et al., 2019), knowledge distillation (Sun et al., 2019; Sanh et al., 2019; Jiao et al., 2019), quantization methods (Zhang et al., 2020; Bai et al., 2020; Kim et al., 2021), and adaptive inference (Zhou et al., 2020; Xin et al., 2020b; Geng et al., 2021; Liu et al., 2020). Early Exit (EE) methods (Teerapittayanon et al., 2016; Zhou et al., 2020; Fei et al., 2022) is one of the adaptive inference methods where intermediate classifiers (exits) are added after every layer. The difficulty of the sample is determined using confidence in the prediction, and the sample is inferred early based on the confidence score exceeding a pre-defined threshold. The confidence score becomes a crucial part of the inference process and decides the sample hardness.

EE strategies either perform confidence-based exiting (Xin et al., 2020b; Bajpai & Hanawal, 2024a) or a patience-based exiting (Zhou et al., 2020) depending on the prediction consistencies treating each classifier equally. Recently EEIC (Sun et al., 2021) decided on exiting based on majority voting between the exits. This method also treats each classifier equally. These methods either consider the confidence of individual exits or utilize the predictions made by exits to define the confidence scores. For instance, in Fig 1, an input sample is currently processed till the third exit; for confidence-based exiting, it checks the confidence at the third exit, ignoring all the information gathered from previous

exits. The patience-based exiting requires predictions of all exits to be consistent if it wants to make an exit. Also, prediction from the first classifier is treated as important as the third, which should not be the case as deeper layers have more information. Similar is the case with majority voting, where all the classifiers are treated equally. They also do not utilize the confidence available from previous layers, thus discarding available information.

This necessitates the requirement of an EE strategy that can utilize the information available at the exits to effectively mitigate the overthinking issue while speeding up the inference. Also, the existing methods do not offer any theoretical viewpoint or make strong assumptions, e.g., all the layers have the same error rate, which makes them less desirable.

We present an EE mechanism named BEEM: Boosting Performance of Early Exit DNNs using Multi-Exit Classifiers as Experts motivated by ensemble learning (Dong et al., 2020), to improve performance of EEDNNs. We treat each intermediate exit classifier as an expert that outputs confidence values on the labels for each input. This confidence score is then weighted based on the expert's accuracy in predictions or the associated prediction cost, i.e., higher weights to deeper exits and vice versa. By treating each exit as an expert, BEEM ensures that the model leverages the strengths of each exit and does not discard the scores of the previous layers if their predictions are in agreement. To determine if a sample can exit at the $i$th classifier, we accumulate weighted confidences of the immediate previous layers whose predictions are in agreement with the $i$th classifier. In case of disagreement with the immediate previous layer, the confidence score resets to the score of the current exit, ignoring past aggregated values. This score is subsequently compared to a predefined threshold for EE decisions.

The exit decision at each layer is based on a cumulative confidence score exceeding a threshold value. The thresholds play a pivotal role as they offer a means to model the trade-off between accuracy and latency. In Section 3.4, we introduce a novel approach to determine the threshold values for different exits by converting the problem of choosing thresholds to a simple linear program. We utilize the error rate of exits to set the threshold values, forcing the exit classifier to perform better than the final classifier of DNNs. In Section 3.5, we also perform a theoretical analysis to derive a condition based on the error rate of the intermediary layer under which BEEM performs better than the vanilla DNN inference.

We experiment with widely adopted Pre-Trained Language Models (PLMs) and encoder-decoder models to perform experiments on GLUE (Wang et al., 2019a) and COCO dataset (Lin et al., 2014). We show that BEEM outperforms all the previous EE methods in terms of speed as well as accuracy. BEEM increases the inference speed by $1.5\times$ - $2.1\times$ with accuracy close to the final layer. For easier NLP tasks such as sentiment analysis, BEEM even outperforms the final layer in terms of accuracy.

In summary, our contributions are as follows:

- We propose new criteria to make EE decisions in DNNs. It combines the confidence score of the intermediary exit classifier to produce an ensemble effect to make PLMs more efficient, robust, and adaptive.

- We provide a method to set threshold values in BEEM by analyzing the error rates of exit classifiers (Section 3.4). This not only helps BEEM achieve better speed-up but also improves accuracy compared to inference at the last layer. We also derive a condition under which this performance is guaranteed.

- Extensive evaluation showed speed-up improvement in both GLUE and COCO datasets. For the GLUE dataset, accuracy also improves with speedup due to a reduction in the overthinking issues of the DNNs.

## 2  RELATED WORK

Early exit methods are applied for various tasks such as image classification, image captioning and NLP tasks to reduce the computational resources and inference latency.

**Early exits in Image tasks:** For image classification tasks, BranchyNet (Teerapittayanon et al., 2016) uses classification entropy at each attached exit to decide whether to infer the sample at the side branch based on the entropy of prediction. Shallow-deep (Kaya et al., 2019) and MSDNet

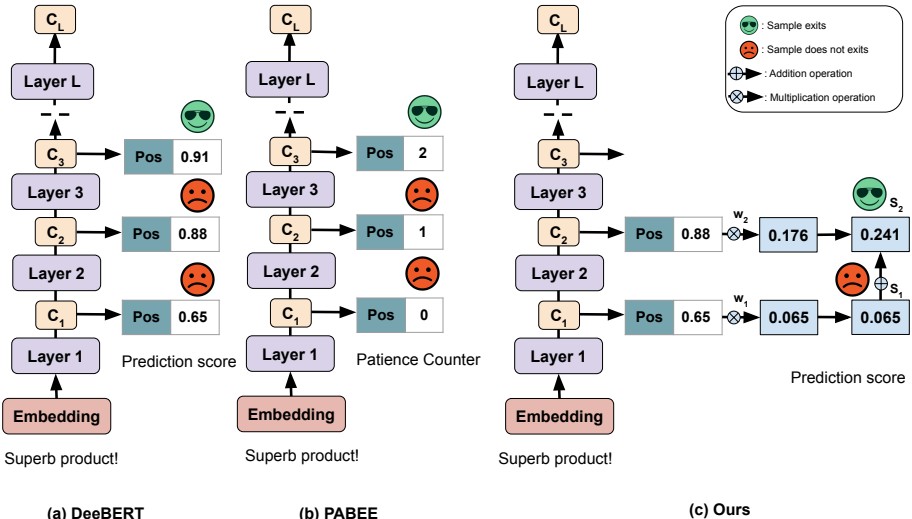

Figure 1: Comparison between (a) DeeBERT, which uses the confidence available at each exit as the metric or deciding early inference (set to 0.9), (b) PABEE, which uses the consistency in prediction as the confidence metric (set to 2) and (c) BEEM that uses the weighted confidence $S_i$ (weights = $[0.1, 0.2, \ldots, 1.2]$) and threshold $\alpha = 0.2$. In BEEM, by appropriately considering information from previous classifiers, a correct prediction is made early which was not the case with others.

(Huang et al., 2017) improve upon BranchyNet by effectively choosing the thresholds based on the confidence distribution. Similar architectures (Pacheco et al., 2021; Dai et al., 2020) split the NN to be deployed on edge and cloud. SEE (Wang et al., 2019c) work in service outage scenarios. FlexDNN (Fang et al., 2020) and Edgent (Li et al., 2019) focus mainly on the most appropriate Neural Network (NN) depth. Other works such as Dynexit (Wang et al., 2019b) focus on deploying the multi-exit NN in hardware. It trains and deploys the NN on Field Programmable Gate Array (FPGA) hardware while Paul *et al.* (Kim & Park, 2020) explains that implementing a multi-exit NN on FPGA board reduces inference time and energy consumption. ZTW (Sun et al., 2021) uses a combination of probability distribution to decide to exit and the combination is learned during training reducing its generalization capabilities. JEI-DNN (Regol et al., 2023) on the other hand uses a gating mechanism for inference where which gate will be opened for a sample is learned during training. In a parallel vein, the MuE, DeeCap and CapEEN (Tang et al., 2023; Fei et al., 2022; Bajpai & Hanawal, 2024b) model employ a distinctive approach to apply early exits to image captioning. DeeCap only applies to the decoder, while MuE applies to the encoder and the decoder. CapEEN makes the exiting process more robust to noisy images.

**Early exit in PLMs:** Multiple approaches have been proposed to effectively apply early exits to PLMs and solve multiple NLP tasks (Liu et al., 2021a; Xin et al., 2020a; Zhou et al., 2020; Banino et al., 2021; Balagansky & Gavrilov, 2022; Sun et al., 2022; Ji et al., 2023). DeeBERT (Xin et al., 2020a), ElasticBERT (Liu et al., 2021a) and BERxiT (Xin et al., 2021). BERxiT proposes an efficient fine-tuning strategy for the BERT model with attached exits. DeeBERT is obtained by training the exit points attached before the last module to the BERT backbone separately. In contrast, ElasticBERT is obtained by training all the exit points attached to the BERT backbone jointly. PABEE (Zhou et al., 2020) is another multi-exit model that makes the exit decision based on the stability of the predictions after different exits. LeeBERT (Zhu, 2021) proposed a self-distillation framework that has similar exiting criteria as PABEE. ETFEE (Ji et al., 2023) adds an adapter on top of the transformer layers and an (Entangled Frame) ETF classifier to make intermediate exits learn better. CeeBERT (Bajpai & Hanawal, 2024a) and DAdEE (Bajpai & Hanawal, 2024c) propose multiple methods to adapt early exits to various domains in an unsupervised setup. Bajpai et al. (2024b;a); Bajpai & Hanawal (2024d) utilize early exits for distributed inference setup.

Our approach differs from past works as 1) Unlike previous studies, BEEM utilizes the ensemble learning principles by treating each exit as an expert. 2) Our work proposes an early exiting method

that utilizes each expert based on its strengths. 3) We also provide a method to set the thresholds using the error rates of the exit classifiers to perform better than the final classifiers.

## 3 PROBLEM SETUP

We start with a pre-trained DNN and attach exit classifiers after each layer. We provide details on training the PLMs for language tasks and the encoder-decoder backbone for image captioning.

### 3.1 PLMs

**Training:** BEEM requires the training of exit classifiers that provide predictions based on their respective layer outputs. Let $\mathcal{D}$ denote the dataset distribution with the label class $\mathcal{C}$ employed for backbone training. Given an input sample $(x, y) \sim \mathcal{D}$, the loss for $i$th exit is calculated as:

$$\mathcal{L}_i(\theta) = \mathcal{L}_{CE}(f_i(x, \theta), y) + KL(p_i, p_L) \tag{1}$$

where $f_i(x, \theta)$ represents the output of the exit attached at the $i$-th layer, $\theta$ denotes the set of all learnable parameters, $\mathcal{L}_{CE}$ denotes the cross-entropy loss and $KL$ is the KL-divergence loss used to additionally train the exits with soft labels from the final layer. KL divergence (Kullback-Leibler divergence) is used in knowledge distillation because it measures how well one probability distribution (the student model's predictions) approximates another (the teacher model's predictions). In the context of knowledge distillation, KL divergence serves as a key component to transfer "soft knowledge" from the teacher to the student. Let $p_i = \mathcal{P}_i(c|x)$ denote the probability distribution over the set of output classes at the $i$th layer, where $\mathcal{P}_i(c|x)$ denotes the estimated probability that $x$ belongs to class $c$. We simultaneously optimize parameters for all exit classifiers, following the methodology proposed by Kaya et al. (2019). The loss function is defined as $\mathcal{L} = \frac{\sum_{i=1}^{L} i \mathcal{L}_i}{\sum_{i=1}^{L} i}$, considering the weighted average to account for the relative inference cost of each exit classifier where $L$ denotes the number of layers in the model. Note that the importance of KL-divergence loss is well-explained in Zhu (2021). Following this training, the model is ready for inference.

**Inference:** We illustrate the inference process of BEEM in Fig. 1. As the input instance $x$ goes through layers $1, 2, \ldots L$ sequentially, the classifier attached to that layer predicts a class label distribution. For $i$th exit classifier, let $C_i$ denote the confidence in the estimate at the $i$th exit. We define $C_i$ as the maximum of the estimated probability class, i.e., $C_i := \max_{c \in \mathcal{C}} \mathcal{P}_i(c|x)$. We denote $\hat{y}_i = \arg\max_{c \in \mathcal{C}} \mathcal{P}_i(c|x)$, the prediction of $i$th exit. Based on the confidence scores we define a weighted confidence score, denoted $S_i$ as:

$$S_i = \begin{cases} S_{i-1} + w_i C_i & \text{if} \quad \hat{y}_{i-1} = \hat{y}_i \\ w_i C_i & \text{if} \quad \hat{y}_{i-1} \neq \hat{y}_i \end{cases} \tag{2}$$

The inference process halts when $S_i \geq \alpha$, where $\alpha$ represents a predefined threshold, and exits with label $\hat{y}_i$. Otherwise, the sample is processed in the next layer and the process completes. If this condition is never met at any exit classifiers, a label is assigned by the classifier at the final layer. This allows a sample to exit the backbone early if the condition is satisfied, avoiding traversal through all layers.

### 3.2 ENCODER-DECODER MODELS

**Encoder-decoder model:** For the image captioning task where the objective is to generate a caption for an input image, we start with a pre-trained encoder and decoder model. We use the Swin Transformer (Liu et al., 2021c) as an encoder and GPT-2 (Radford et al., 2019) as a decoder. We attach exits to the decoder of the backbone. The backbone is trained using cross entropy and KL-divergence loss where loss for $i$th exit could be written as:

$$\mathcal{L}_i(\theta) = \sum_{t=1}^{T} (\mathcal{L}_{CE}(f_i(x, \theta, y_{1:t-1}), y_t) + KL(p_t^i, p_t^L)),$$

where $\theta$ is the collection of all the parameters, $x$ is the input image, $T$ is the caption length, $y_{1:T}$ is the ground-truth caption. $p_t^i$ is the probability vector on the vocabulary $\mathcal{V}$ for $i$th exit. Its $v$th

component could be written as $p_t^i(v) = \mathcal{P}_i(v|y_{1:t-1}, x; \theta)$ where $\mathcal{P}_i$ is the probability distribution output over $\mathcal{V}$ by the $i$th exit. Note that $L$ here is the number of layers in the decoder. Similarly, we define $p_t^L$ as the probability vector for the final decoder layer. The overall loss across all the exits is the same as for the PLM training.

**Caption inference:** We predict the caption in an autoregressive manner. This entails making a token-by-token prediction for a given image. In this case, the confidence could be formulated as $C_i = \max_{v \in \mathcal{V}} \mathcal{P}_i(v|\hat{y}_{1:t-1}, x; \theta)$ where $\mathcal{P}_i$ is same as defined above and $\hat{y}_{1:t-1}$ is the predicted caption till $(t-1)$th word. A token will be predicted at which exit is decided by equation 2. For an input image $x$, we start the caption with begin of the sentence token and then the inference process stops after the end of the sentence token is predicted.

In Figure 1, confidence-based early exit methods like DeeBERT and ElasticBERT, relying on softmax scores, tend to be overly confident toward a single class and classifier. Such methods also face the consequences of ignoring information obtained from previous classifiers as they progress to the next one. This limitation is addressed by patience-based approaches like PABEE, which decide to exit when predictions show consistency across multiple classifiers. However, patience-based methods treat each classifier equally and underutilize valuable information available in terms of prediction scores, which affects the adaptability of the model.

Contrarily, BEEM captures the confidence available at each exit and assigns weights to each classifier based on its accuracy or cost. It takes into account the consistency in predictions by reducing the score to the current classifier's weighted confidence when predictions are inconsistent. This unique approach in BEEM incorporates both patience and confidence to make predictions. Note that the confidence score $S_i$ given in equation 2 can predict hard samples early as if the predictions of initial classifiers are consistent but with low confidence, the summed-up $S_i$ score makes them exit early as shown in Figure 1. Also, it effectively mitigates errors arising from a single classifier while considering the confidence in predictions and weighing them based on their performance.

### 3.3 ASSIGNING WEIGHTS

In this section, we provide methods to set the weights for the exits.

1) **Cost vector:** First, we consider weights as the cost of getting inference from the exit classifier where the cost could be in the form of $w_i = \lambda i$ where $\lambda$ is the processing cost of the one exit and since the layers are identical, the cost is a multiple of $\lambda$ for deeper layers.

2) **Accuracy:** We can also consider the weights as the accuracy of each classifier. The accuracy could be calculated on a validation dataset. This will provide weights to exits depending on how much accurate a particular expert (exit) is.

Note that the major difference between the existing methods is the cost-based weights have a task to reduce the overall cost while sacrificing some accuracy while the accuracy-based methods will focus more on accuracy. Note that using accuracy-based weights can also improve the efficiency that comes because of overthinking issues. As in the accuracy-based, we know the true capability of each exit.

### 3.4 CHOICE OF THRESHOLDS $\alpha$

We can choose the threshold values in two ways, one way is to choose the best-performing threshold on the validation set, and the other is based on forcing error rates to be smaller than the error rate of the final classifier.

1) **Classical method:** We choose the search space for threshold $\alpha \in S = \{0.3, 0.6, 0.9, 1.2, 1.5\}$. The values of $w_i \in [0, 1]$, $C_i \in [0, 1]$ imply that $S_i \leq L$ i.e. the score at any exit layer $i$ cannot be greater than the number of layers $L$ as $S_i$ is a multiple of two values between 0 and 1, it becomes very small and is added almost $L$ times. We choose the best-performing threshold on the validation set in terms of accuracy.

2) **Using error rates:** Let us consider that $c_{misc}^t$ represents the number of samples that exit at $t$th classifier with a misclassification, $c_{stop}^t$ represents the number of samples that exit at the $t$th classifier. Note the $c_{stop} = \sum_{j=1}^{n} \mathbb{1}_{\{C_j \geq \alpha\}}$ can also be considered as the coverage of the $t$th classifier and

$c_{misc}^t = \frac{\sum_{j=1}^n \mathbb{1}_{\{\hat{y}_j \neq y | C_j \geq \alpha\}}}{c_{stop}}$, where $n$ is the total number of samples in the dataset. $p$ is the error rate of the final classifier, then we observe that our algorithm will perform better than final layer if $c_{misc}^t / c_{stop}^t < p$ for every exit classifier $t$. The above condition tells that the fraction of the samples that have exited and misclassified (i.e., error rate) at $t$th classifier should be less than the error rate of the final classifier. If the above condition is satisfied for all the exits then we are guaranteed that BEEM can outperform the final classifier of the PLM. The objective is to maximize speedup while satisfying the above condition.

Note that the error rate depends on the threshold $\alpha$, a higher value of the threshold will lower the error rate as then samples with higher confidence will exit reducing the chance of misclassification. Observe that, we can find the threshold value $\alpha_t$ for the $t$th classifier such that the condition $c_{misc}^t / c_{stop}^t < p$ is satisfied on the validation set. We define $q_{\alpha_t}$ as the error rate associated with the threshold $\alpha_t$ for the $t$th classifier. We can set the threshold by solving the optimization problem.

$$\begin{aligned} \underset{\alpha_t \in S}{\text{minimize}} \quad & \alpha_t \\ \text{subject to} \quad & q_{\alpha_t} \leq p, \end{aligned} \quad (3)$$

where the set $S = \{0.5, 1, \ldots, 5, L\}$ is the search space for the thresholds. Note that we have added $L$ in the search space so that the problem always remains feasible. By solving Eq. 3, our method finds the optimal threshold that has maximum speedup while performing better than the final layer. Also, observe that the above problem has small computational complexity as the minimization is over a very small finite set.

## 3.5 THEORETICAL ANALYSIS

**Theorem 3.1.** *Consider an early exit PLM with $L$ layers. Let $p$ denote the error rate of the final classifier and the error probability of $i$th exit classifiers be $q_i$ such that $q_i < \frac{a_i}{a_i + ((1/p - 1)b_i^{i-1})}$ holds for all exit layers $i = 1, 2, \ldots, L-1$ where $a_i$ and $b_i$ are constants for a given exit $i$. Then, the error probability of BEEM is better than $p$ i.e., it performs better than the final layer.*

The proof of the theorem is given in the Appendix A.1. Note that the above theorem proves the general condition for better performance of BEEM and does not depend on the threshold values $\alpha$. $a_i$ denotes the ratio of the probability of exiting with one change in prediction to the probability of exiting with zero changes in prediction till $i$th classifier and $b_i = \frac{q_i^{max}}{q_i^{min}}$ where $q_i^{max} = \max\{q_1, \ldots, q_i\}$ while $q_i^{min} = \min\{q_1, \ldots, q_i\}$. Observe that as we move deeper into the backbone the bound becomes tighter which makes sense as deeper layers are more likely to be accurate. Also, the bound is inversely proportional to the error rate of the final layer, if the error rate of the final layer is smaller, then the bound gets tighter. Previous method Zhou et al. (2020) had a very strong assumption while providing a similar condition for their method, it assumed that all the classifiers have the same error rate which is not true. If we impose the same condition the bound simplifies to $q_i < \frac{a_i}{a_i + ((1/p - 1))}$ which is a more simplified and stronger bound than PABEE (Zhou et al., 2020).

## 4 EXPERIMENTS

In this section, we provide the details of the experiments performed in this work.

**Datasets:** We evaluate our approach using the GLUE benchmark datasets (Wang et al., 2019a). Our assessments encompass diverse tasks, such as sentiment classification using the Stanford Sentiment Treebank (SST-2), Natural Language Inference (NLI) tasks with Multi-Genre Natural Language Inference (MNLI), Question Natural Language Inference (QNLI), and Recognizing Textual Entailment (RTE). For Paraphrase Similarity Matching, we include Microsoft Research Paraphrase Matching (MRPC) and Quora Question Pairs (QQP), while Linguistic Acceptability is measured using The Corpus of Linguistic Acceptability (CoLA). In instances where datasets comprise multiple units, we report the arithmetic mean. We exclude the WNLI task, following previous works (Devlin et al., 2018; Zhu, 2021; Zhou et al., 2020). For captioning, we use the COCO (Lin et al., 2014) dataset.

**Baselines:** We compare against the vanilla DNN exiting and other techniques that speed up DNN inference. The baselines are as follows:

| Model/Data | SST-2 | | MNLI | | RTE | | QNLI | | QQP | |
|---|---|---|---|---|---|---|---|---|---|---|
| | Acc | Speed | Acc | Speed | Acc | Speed | Acc | Speed | Acc | Speed |
| *Dev set* | | | | | | | | | | |
| ALBERT | 92.4 | 1.00x | 84.5 | 1.00x | 77.9 | 1.00x | 91.3 | 1.00x | 90.6 | 1.00x |
| ALBERT-9L | -1.6 | 1.33x | -3.2 | 1.33x | -2.5 | 1.33x | -2.7 | 1.33x | -1.5 | 1.33x |
| DeeBERT | -2.3 | 1.72x | -2.9 | 1.65x | -3.1 | 1.78x | -1.9 | 1.57x | -2.5 | 1.81x |
| ElasticBERT | -2.1 | 1.75x | -2.3 | 1.71x | -2.7 | 1.81x | -1.7 | 1.66x | -2.1 | 1.78x |
| FastBERT | -1.1 | 1.85x | -0.3 | 1.61x | -0.2 | 1.79x | -0.8 | 1.71x | -0.3 | 1.88x |
| PABEE | -0.1 | 1.87x | -0.5 | 1.85x | -0.7 | 1.64x | -0.6 | 1.81x | -0.2 | 1.68x |
| ZTW | -0.2 | 1.64x | -0.3 | 1.67x | +0.2 | 1.63x | -0.3 | 1.75x | -0.1 | 1.71x |
| PCEEBERT | +0.1 | 1.24x | 0.0 | 1.31x | +0.3 | 1.27x | -0.1 | 1.21x | +0.1 | 1.37x |
| LeeBERT | 0.0 | 1.78x | -0.2 | 1.74x | -0.1 | 1.59x | +0.1 | 1.79x | -0.2 | 1.97x |
| PALBERT | -0.4 | 1.54x | -0.8 | 1.61x | +0.3 | 1.45x | -0.2 | 1.59x | -0.1 | 1.63x |
| JEI-DNN | -0.1 | 1.77x | +0.1 | 1.67x | 0.0 | 1.35x | -0.1 | 1.43x | +0.2 | 1.57x |
| BEEM-C | 0.0 | 1.71x | +0.1 | **2.03x** | +0.4 | 1.79x | 0.0 | 1.90x | 0.0 | 1.93x |
| BEEM-A | **+0.4** | **1.98x** | **+0.3** | 1.96x | **+0.7** | **1.89x** | **+0.2** | **1.92x** | **+0.5** | **2.09x** |
| *Test set* | | | | | | | | | | |
| ALBERT | 92.3 | 1.00x | 84.2 | 1.00x | 72.1 | 1.00x | 90.9 | 1.00x | 80.1 | 1.00x |
| ZTW | -0.4 | 1.61x | -0.5 | 1.52x | +0.1 | 1.64x | -0.1 | 1.59x | -0.5 | 1.81x |
| LeeBERT | -0.5 | 1.79x | -0.9 | 1.88x | 0.0 | 1.68x | -0.4 | 1.72x | -0.3 | 1.86x |
| PALBERT | -0.3 | 1.49x | -1.1 | 1.72x | +0.2 | 1.27x | -0.4 | 1.51x | -0.3 | 1.50x |
| JEI-DNN | -0.1 | 1.35x | -0.7 | 1.59x | 0.0 | 1.36x | -0.2 | 1.39x | 0.0 | 1.47x |
| BEEM-C | -0.2 | **1.98x** | -0.4 | 1.95x | +0.1 | 1.74x | +0.1 | 1.81x | +0.1 | **1.97x** |
| BEEM-A | **+0.4** | 1.91x | **-0.3** | **2.06x** | **+0.6** | 1.77x | **+0.5** | **1.88x** | **+0.2** | 1.95x |

Table 1: Main results: This table compares BEEM against all the state-of-the-art early exiting baselines. We report the accuracy (Acc in %) and Speed-up (Speed).

**1) Final layer:** The final layer of the DNN model, referred to as the "final layer" in Table 1.

**2) Reducing layers:** We use only the first 9 layers of the DNN model with a single output layer, denoted as DNN-9L. This serves as a performance lower bound since it employs no EE techniques.

**3) Early-exit models:** DeeBERT (Xin et al., 2020b) and ElasticBERT (Liu et al., 2021b): Use fixed confidence thresholds for early exits. FastBERT (Liu et al., 2020): Uses a self-distillation framework to train intermediate exits. PABEE (Zhou et al., 2020) and LeeBERT (Zhu, 2021): uses prediction stability, with LeeBERT incorporating knowledge distillation. ZTW (Sun et al., 2021): combines the output probability outputs across all the layers and trains the weights provided as additional parameters for training. PCEEBERT (Zhang et al., 2022): Combines confidence and patience metrics, similar to PABEE. MuE (Tang et al., 2023): Uses hidden representation similarity for early exits, applied to the BERT-base model. PALBERT model (Balagansky & Gavrilov, 2022): State-of-the-art methods that face adaptation challenges due to training dataset bias. PALBERT uses Lambda layers (Banino et al., 2021). JEI-DNN (Regol et al., 2023) performs exiting using a gating mechanism where it learns a probability distribution over all exits and decidesto exitg based on that. DeeCAP (Fei et al., 2022) is specifically for image captioning that uses an imitation network to mimic the behaviour of the decoder model.

We utilized the codebases of existing methods to get the results, all the results were obtained using the hyperparameters given in their available codes. Note that for the encoder-decoder model, we extend the ideas of DeeBERT, FastBERT, PABEE, and LeeBERT to the decoder of the backbone.

## 4.1 EXPERIMENTAL SETUP

Our experiments are conducted on a single NVIDIA RTX 2070 GPU, The runtimes are given below.

**Training.** For the training phase, we augment the pre-trained BERT/ALBERT model with a linear output layer after each intermediate layer to serve as an exit point. We conduct a grid search over batch sizes of $\{8, 16, 32\}$ and learning rates $\{2e\text{-}5, 3e\text{-}5, 4e\text{-}5, 5e\text{-}5\}$ using the Adam (Kingma & Ba, 2014) optimizer.

Incorporating an early-stopping mechanism, we select the best model based on the validation set. These parameters are fixed to 16 batch size and 3e-5 learning rate for the encoder-decoder backbone.

| Model/Data | RTE | | CoLA | |
|---|---|---|---|---|
| | Acc | Speed | Acc | Speed |
| BERT | 69.3 | 1.00x | 57.8 | 1.00x |
| BERT-9L | -1.8 | 1.33x | -2.1 | 1.33x |
| DeeBERT | -2.5 | 1.47x | -1.5 | 1.21x |
| ElasticBERT | -2.2 | 1.52x | -1.2 | 1.18x |
| FastBERT | -0.8 | 1.44x | -0.2 | 1.24x |
| PABEE | -1.1 | 1.62x | -0.1 | 1.16x |
| ZTW | -0.7 | 1.52x | -0.5 | 1.48x |
| LeeBERT | -0.6 | 1.60x | -0.1 | 1.28x |
| PALBERT | -0.5 | 1.32x | -0.6 | 1.19x |
| JEI-DNN | -0.2 | 1.30x | -0.3 | 1.18x |
| BEEM-C | -0.1 | 1.63x | +0.0 | 1.30x |
| BEEM-A | **+0.2** | **1.70x** | **+0.3** | **1.49x** |

Table 2: Results on the BERT backbone on the GLUE datasets. We report accuracy (in %).

| Models/Metric | BLEU-1 | BLEU-4 | METEOR | CIDEr | SPICE | ROUGE-L | Speedup |
|---|---|---|---|---|---|---|---|
| Final-Exit | 82.5 | 42.3 | 32.2 | 147.1 | 26.7 | 61.3 | 1.00x |
| Decoder-9L | 76.5 | 37.1 | 29.3 | 134.8 | 23.2 | 57.9 | 1.33x |
| DeeBERT | 70.1 | 32.3 | 26.9 | 110.2 | 20.9 | 50.7 | 1.35x |
| ElasticBERT | 71.4 | 32.8 | 27.6 | 114.6 | 21.4 | 51.6 | 1.37x |
| PABEE | 72.7 | 33.9 | 27.9 | 115.6 | 21.9 | 52.3 | 1.30x |
| FastBERT | 75.0 | 35.6 | 28.2 | 119.5 | 22.1 | 53.7 | 1.42x |
| LeeBERT | 77.3 | 38.7 | 29.4 | 129.2 | 23.0 | 55.9 | 1.39x |
| DeeCap | 77.5 | 39.2 | 29.9 | 132.8 | 23.2 | 56.9 | 1.60x |
| MuE | 79.3 | 40.5 | 30.9 | 139.4 | 24.9 | 59.7 | 1.64x |
| **BEEM-C** | 81.8 | 41.5 | 31.7 | 145.1 | 25.9 | 60.1 | **1.71x** |
| **BEEM-A** | **82.4** | **42.1** | **32.0** | **146.5** | **26.3** | **60.9** | 1.67x |

Table 3: Results showing that BEEM outperforms the other baselines on test split of COCO dataset.

The training time has an average GPU runtime of around 10 hours on a dataset, with the COCO dataset exhibiting the highest runtime ($\sim$ 26 hours).

**Inference:** Following the previous methodology on input-adaptive inference (Teerapittayanon et al., 2016; Kaya et al., 2019), the inference is performed on a per-instance basis, setting the batch size to 1. This aligns with scenarios where low latency is critical, such as processing individual requests from different users (Schwartz et al., 2020). The reported values represent the median results from 5 runs with different seeds as small datasets such as CoLA and RTE have high variance in performance. For performing inference, the average runtime was $< 20$ minutes for NLP datasets. For COCO dataset the runtime was 5 hours on the Karpathy test split.

**Metric.** We report the speed-up ratio as a metric for measuring time reduction to remain consistent with the previous methods. Speed-up could be defined as: $\frac{\sum_{i=1}^{L} L \times n_i}{\sum_{i=1}^{L} i \times n_i}$ where $n_i$ are the number of samples exiting from the $i$th layer. For the image captioning task $n_i$ is the number of words exiting from the $i$th layer. This metric could be interpreted as the increase in speed of the model as compared to the naive (AL)BERT model. This metric can be converted to expected time reduction rate.

In Table 1 and 2, we present results wherein classifiers are assigned weights based on the cost of each classifier denoted as BEEM-C and where the weights are set using the accuracy on the validation set, we denote it by BEEM-A.

## 5 RESULTS

In this section, we highlight and discuss the key findings of our work. Tables 1 and 2 present the results when ALBERT and BERT serve as the backbone models, respectively. BEEM consistently outperforms all previous baselines by a significant margin. A major observation is a notable enhancement in BEEM as compared to the performance of (AL)BERT models, except for a minor setback on the MNLI dataset. The improvement in accuracy by BEEM may be attributed to the thresholds being chosen after solving the constraint optimization 3 exclusively on the validation

dataset. Table 3 shows results on the COCO dataset and observes significant improvements by using BEEM.

The substantial accuracy drop observed in DeeBERT and ElasticBERT results from a direct comparison with entropy, neglecting the information utilized by preceding classifiers. Conversely, PABEE, LeeBERT, FastBERT, and ETFEE employ patience-based early exit criteria, posing a stringent criterion for exiting. ZTW is one of the works that weights the classifier and utilizes the ensemble techniques but suffers from poo generalization as weights are learned restricting better generalization as well as adding complexity. JEI-DNN uses the gating mechanism to decide whether to exit and does not utilize the information of multiple available classifiers. Similar is the case with DeeCAP and MuE, and for image captioning, they do not perform any knowledge distillation, further reducing the performance. These methods do not account for the confidence available at each exit, assigning them equal weight irrespective of their varying confidence level prediction. This lack of consideration impacts the adaptiveness of early exit models.

BEEM-C and BEEM-A are the two variants of our proposed method. In the results, we can observe that BEEM-A consistently outperforms BEEM-C for all the datasets in terms of accuracy and most of the datasets in terms of speed-up. This gain for BEEM-A could be attributed to the assumption in BEEM-C that the cost of exits (experts) was set by assuming that it is directly proportional to the accuracy but this is not true due to the overthinking issue. Still BEEM-C performs better than previous baselines on the datasets in which the overthinking issue is minimal. The main advantage of BEEM-C is that since the thresholds are fixed in our setup, we can still tune the cost $\lambda$ based on the speed-up (see section A.3) needed which is unavailable in BEEM-A.

## 6 ABLATION STUDY AND ANALYSIS

In this section, we provide the results of our method on (AL)BERT large models. In the Appendix, we perform an analysis of the behaviour of parameters $\alpha$ and $\lambda$ (see Appendix A.2, A.3). This analysis shows that our methods not only have better performance but also better models for the accuracy-efficiency trade-off i.e., the drop in accuracy of BEEM was lower when speedup increases as compared to others.

### 6.1 PLM SIZE

In Table 4, we analyze BEEM's performance on ALBERT-Large models, each with 24 layers.

Our results show a significant acceleration in processing speed, especially for larger models, due to their inherent overparameterization. This efficiency gain underscores BEEM's potential for optimizing large architectures.

| Data | RTE | | CoLA | | QQP | |
|---|---|---|---|---|---|---|
| | Acc | Spd | Acc | Spd | Acc | Spd |
| AB-L | 80.5 | 1.00x | 60.9 | 1.00x | 91.1 | 1.00x |
| Our-A | +1.8 | 2.04x | +1.3 | 2.85x | +0.1 | 3.33x |
| B-L | 70.9 | 1.00x | 64.3 | 1.00x | 91.2 | 1.00x |
| Our-A | +0.5 | 1.81x | +0.9 | 1.71x | +0.3 | 2.51x |

Table 4: This table provides results on the large variants of (AL)BERT models compared with BEEM-A. AB-L is ALBERT-Large and B-L is BERT-Large.

Furthermore, BEEM notably improves accuracy by mitigating overthinking, where models focus on irrelevant features. This issue is more pronounced in larger models, making BEEM particularly effective. Our findings demonstrate that BEEM enhances performance and speedup for large-scale transformer-based PLMs, becoming increasingly effective with larger model sizes.

### 6.2 CHOICE OF THRESHOLDS

In table 5, we compare results when the thresholds are chosen based on the equation 3 and when the thresholds are set using the vanilla method i.e. best-performing on the validation set. We can observe that, there is a significant increase in the performance in both ALBERT models attributed to the choice of thresholds made by equation 3. Observe that setting thresholds by solving the equation can improve both speedup as well as accuracy, this is as the equation finds the smallest threshold

| Method | | SST-2 | | QQP | |
|---|---|---|---|---|---|
| Our method | | Acc | Spd | Acc | Spd |
| Base | w/o fix | 92.4 | 1.81x | 90.4 | 2.05x |
| | w fix | 92.6 | 1.98x | 91.1 | 2.09x |
| Large | w/o fix | 93.1 | 2.19x | 91.1 | 2.95x |
| | w fix | 93.4 | 2.31x | 91.3 | 3.33x |

Table 5: This table compares the setting of thresholds based on the best-performing threshold on the validation dataset (w/o fix) and fixing the threshold after solving equation 3 (w fix) on ALBERT-Base/Large models.

that can improve the accuracy from the final layer. The thresholds are set such that each of them perform equivalent to the final layer.

## 7 CONCLUSION

In conclusion, our study introduces a novel framework BEEM designed to enhance the efficiency, robustness, and adaptability of early exiting strategies in DNNs. By leveraging multiple exit classifiers, where each exit is treated as an 'expert', and their outputs are combined to create an ensemble effect. Our approach considers both prediction confidence and patience, leading to improved performance and reduced latency, particularly advantageous in scenarios with strict latency requirements. Additionally, we propose a method for threshold selection, further enhancing the effectiveness of our approach. We also perform theoretical analysis to provide deep insights into our method. We experimentally validate that the speed-up observed was $1.5\times$ - $2.1\times$ for various NLP and image captioning tasks.

## 8 LIMITATIONS

While the performance of our method is better than the final layer for NLP tasks, it takes a hit for difficult tasks such as image captioning. It happens as the thresholds are being set on the validation dataset that might not generalize well on the test dataset i.e., the solution to the optimization problem 3 might not work for the test dataset. However, as our objective is to minimize the performance loss, BEEM effectively does that and performs better than all the existing early exit models and is comparable to the final layer of DNNs with a large improvement in inference speed.

### ACKNOWLEDGEMENTS

Divya Jyoti Bajpai is supported by the Prime Minister's Research Fellowship (PMRF), Govt. of India. Manjesh K. Hanawal thanks funding support from SERB, Govt. of India, through the Core Research Grant (CRG/2022/008807) and MATRICS grant (MTR/2021/000645), and DST-Inria Targeted Programme. We also thank funding support from Amazon IIT-Bombay AI-ML Initiative (AIAIMLI).

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

## A  APPENDIX

### A.1  PROOF OF THEOREM 3.1

**Proof.** For simplicity, we prove it for the binary classification case. For the samples that are not inferred at intermediate exits, the misclassification probability will remain the same with or without BEEM. Therefore, we only need to consider the case when the sample exits early and misclassified. We denote the probability that the sample exits early from exit $t$ as $p_t^{stop}$ and the probability that the exiting sample is misclassified as $p_t^{misc}$. We next look for conditions under which $\frac{p_t^{misc}}{p_t^{stop}} < p$, i.e., the fraction of samples early exiting being misclassified is less the error rate of the final classifier.

Let us consider two random variables $X_t$ and $Y_t$ where $X_t = 1$ if the sample exits at $t$th classifier else 0 and $Y_t = 1$ if the prediction at $t$th classifier is correct else 0. Now the probability that a sample exits at the $t$th classifier could be written as:

$$P(X_t = 1) = P(S_1 < \alpha, \ldots S_{t-1} < \alpha, S_t \geq \alpha) \tag{4}$$

Let $q_t^{min} = \min\{q_1, q_2, \ldots, q_{t-1}\}$ and $q_t^{max} = \{q_1, q_2, \ldots, q_{t-1}\}$. We denote the prediction of classifier $t$ as $\hat{y}_t$. Note that $P(y^* = \hat{y}_t) = 1 - q_t$. We define a counter $Y_t$ as:

$$\mathbf{c}_t = \begin{cases} 1 & if \quad \hat{y}_{t-1} = \hat{y}_t \\ 0 & if \quad \hat{y}_{t-1} \neq \hat{y}_t \end{cases} \tag{5}$$

We define $\mathbf{C}_t = \sum_{i=1}^{t} \mathbf{c}_i$. Note that $\mathbf{C}_t$ monitors the number of times the prediction has been changed till the $t$th classifier.

$$p_t^{stop} = P(X_t = 1) =$$

$$\sum_{c=0}^{t} P(X_t = 1 | \mathbf{C}_t = c) P(\mathbf{C}_t = c) \tag{6}$$

We denote $P(X_t = 1 | \mathbf{C}_t = c)$ as $A_t^c$. We have

$$p_t^{stop} = P(X_t = 1) = \sum_{c=0}^{t} A_t^c P(\mathbf{C}_t = c).$$

By Total Probability law, we can say that

$$P(X_t = 1) = P(X_t = 1 | mc) P(mc)$$

$$+ P(X_t = 1 | cc) P(cc) \tag{7}$$

where $mc$ is misclassified and $cc$ is correctly classified. The ratio is now:

$$\frac{p_t^{misc}}{p_t^{stop}} =$$

$$\frac{P(X_t = 1|mc)P(mc)}{P(X_t = 1|mc)P(mc) + P(X_t = 1|cc)P(cc)} < p \quad (8)$$

Simplifying this, we have,

$$\frac{P(X_t = 1|cc).P(cc)}{P(X_t = 1|mc).P(mc)} > \frac{1}{p} - 1$$

For $P(X_t = 1|mc)$, from the definition of the confidence score 5, we observe that the highest probability of exiting at $t$th layer with misclassification is if all the previous predictions are consistent, i.e., $S_t$ will be highest when all the previous exits misclassified. Hence we have

$$P(X_t = 1|mc) < tA_t^0 q_1 q_2 \ldots q_t < tA_t^0 (q_t^{max})^{t-1} q_t$$

For $P(X_t = 1|cc)$, we observe that again by definition of the confidence score, we can lower bound it, since the lowest probability of exiting with correct classification at $t$th classifier will be when all the previous classifiers had a misclassification and then at $t$th classifier the prediction reversed. Hence, we have that

$$P(X_t = 1|cc) >$$

$$tA_t^1 q_1 q_2 \ldots (1 - q_t) > tA_t^1 (q_t^{min})^{t-1} (1 - q_t) \quad (9)$$

Now we have inequality as

$$\frac{tA_t^1 q_{min}^{t-1}(1 - q_t)}{tA_t^0 q_{max}^{t-1} q_t} > \frac{1}{p} - 1$$

We get the desired result by simplifying the above inequality for $q_t$. We denote the constant term $\frac{A_t^1}{A_t^0}$ as $a_t$. Hence, we finally show with $b_t = \frac{q_t^{max}}{q_t^{min}}$:

$$q_t < \frac{a_t}{a_t + ((1/p - 1)b_t^{t-1})} \quad (10)$$

where $a_t$ and $b_t$ are constant for each $t$.

This concludes the proof.

A.2 ANALYSIS OF THRESHOLDS $\alpha$

In table 5, we compare BEEM-A where thresholds are set by optimizing Eq 3 against the case where threshold values are chosen using the validation dataset and are constant for all exits. We provide the comparison on (AL)BERT-base as well as large models. We can observe a significant difference in accuracy and speedup. The improvement in accuracy and speedup is explained by the formulation of the optimization problem in Sec. 3.4.

Also, in Fig 2a, we plot the accuracy-speedup trade-off curves. We can observe that the change rate of decrease of the accuracy of BEEM by increasing speed-up is smaller as compared to previous baselines. This extra stability by BEEM could be attributed to its characteristic of confirming the predictions from multiple exits (experts). Note that Fig 2a does not assume the thresholds are fixed and varies them to show the stability of the approach.

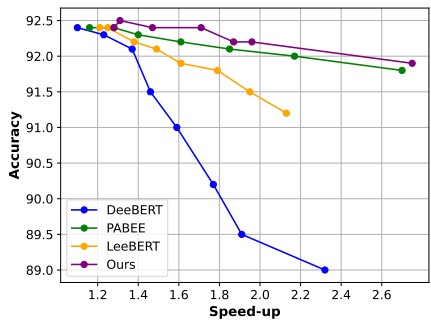

(a) Accuracy-speedup trade-off for $\alpha$.

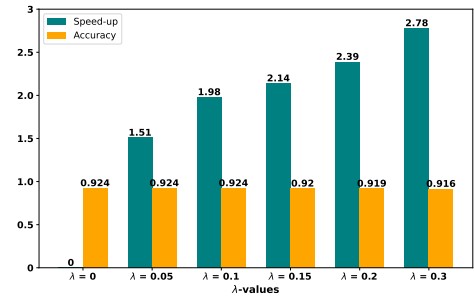

(b) Trade-off by varying $\lambda$.

### A.3 ANALYSIS OF COST $\lambda$ AND # PARAMETERS

In BEEM-C, we introduce weights representing costs associated with utilizing exits. Figure 2b illustrates how altering these costs influences accuracy and speed-up trade-offs. As the cost attributed to each exit classifier increases, we observe a slight decline in accuracy accompanied by a more significant enhancement in speed-up. This hyperparameter thus offers a mechanism for modeling the trade-off between accuracy and speed-up, particularly as the thresholds $\alpha$ remain fixed. It's worth noting that adjusting the value of $\lambda$ impacts the quantity $q_{\alpha^t}$ defined in Section 3.4. Consequently, modifications to $\lambda$ may induce changes in the values of $\alpha^t$.

Note that BERT-base/Large has 110/340 Million Parameters with exits and ALBERT-base/Large has 13/19 Million parameters with exits.

