# OpenReview forum: "BEEM: Boosting Performance of Early Exit DNNs using Multi-Exit Classifiers as Experts"
_ICLR.cc/2025/Conference — ICLR 2025 Poster_

### Official Review · Reviewer_mL6L · 2024-10-17

**Soundness:** 4
**Presentation:** 3
**Contribution:** 3
**Rating:** 8
**Confidence:** 3

**Summary:**

BEEM presents an approach for reducing the latency of model inference by allowing classification decisions to be made at multiple points along the model, with early termination occurring if classification confidence exceeds a threshold. Instead of treating each early exit point independently, BEEM also considers a consensus along a span of exit points to be sufficient for early exiting, even if the confidence of the current exit point is below the threshold that would allow exit by itself.

**Strengths:**

Treating early exiting as an ensemble based consensus mechanism is a good idea, and weighting those exit classifiers based on their ability to classify effectively is also a good idea. As best I can tell, the approach does not introduce hyperparameters without providing a rigorous way to arrive at them, which I think helps with the general utility of the approach. Given that classifiers and next-token generation are major parts of modern AI, and this approach works on both, I think this work has significance.

**Weaknesses:**

I don't feel like there are any showstopping weaknesses that call into question the acceptability of the paper in general. So I will limit it to some constructive feedback, and then hopefully the authors can speak to my questions below.

Figure 1:
- I found this figure a little bit hard to parse, even after reading the rest of the paper. I think that if you had an arrow from the 0.065 box to the addition operator above it, it would be much more obvious what’s happening.

Section 3.4.2:
- Line 262.5: Should it be c_stop or c_stop^t?
- Should c_misc^t be divided by c_stop on line 264, as it’s meant to be the count (“represents the number of samples …”), and thus `c_misc^t / c_stop^t` would be the `p^t` error rate? Otherwise we’re looking at `c_misc^t / c_stop^{2 (t)}` which doesn’t seem to be right.

Section 4:
- It perhaps would be nice to see the early exit behaviors of some of the techniques with a histogram over exit points, for example. Or even simply the average exit layer over each dataset.

Section A.2
- Broken table link on line 782

Nit:
- Wrong parenthesis form on line 319
- If you have space, it might be good to introduce what BEEM-A and BEEM-C mean in Table 1, as it doesn’t show up until the end of section 4.1.Metric.
- Line 445 “poo [sic]” typo

**Questions:**

Equation 2:
Why do we only track the max prediction chain? Alternatively, could we just continue to aggregate weighted confidences until one of them reaches threshold? For example, if the early parts of the model are trying to decide between two classes, and thus have comparable (but oscillating max) confidence, in BEEM, these oscillations cause the S_i score to stay small, and thus requiring a deeper traversal once the two classes are resolved into one (as we need to build confidence for the single resolved class).

Section 4:
Why was DNN-9L chosen? Probably a minor question, but unless that comes from a related work, I'm not sure where the 9 came from.

General:
It seems intuitive that perhaps some layers are only specifically good at EE-classifying a subset of classes. Have you observed any behavior like this? It would tie into my question about equation 2, which is that because you're relying on a chain of consistent max labels, that uneven error distributions could be harming your overall approach.

---

> ### Author Response · Authors · 2024-11-20
> **Rebuttal**
>
> Thanks for the positive remarks and constructive feedback.
>
> Que1 : Line 262.5: Should it be c_stop or c_stop^t?
> Should c_misc^t be divided by c_stop on line 264, as it’s meant to be the count (“represents the number of samples …”), and thus c_misc^t / c_stop^t would be the p^t error rate? Otherwise we’re looking at c_misc^t / c_stop^{2 (t)} which doesn’t seem to be right.
>
> Ans 1: Thanks for pointing it out, yes this is a typo and we will fix that in the final version.
>
>
>
> Que-2: It perhaps would be nice to see the early exit behaviors of some of the techniques with a histogram over exit points, for example. Or even simply the average exit layer over each dataset.
>
> Ans-2: Sure, we will add a histogram with number of exiting samples at different layers in the Appendix. Also, please note that we can find the average layer required for a dataset by reversing the speedup and multiplying it by number of layers in the backbone. Speedup is a metric that can easily be converted to different metrics such as expected time taken, expected time reduction rate and average number of layers required.
>
>
> Que-3: Equation 2: Why do we only track the max prediction chain? Alternatively, could we just continue to aggregate weighted confidences until one of them reaches threshold? For example, if the early parts of the model are trying to decide between two classes, and thus have comparable (but oscillating max) confidence, in BEEM, these oscillations cause the S_i score to stay small, and thus requiring a deeper traversal once the two classes are resolved into one (as we need to build confidence for the single resolved class).
>
>
> Ans-3: Yes that could also be one possibility, but the reason why we do not want to aggregate the confidence until the prediction is consistent is that the model can gain fake confidence for hard samples due to not so good features extracted at initial layers. Hence until the model is consistent in its predictions, we do not want to aggregate the confidence. This reduces the chances of fake confidence being aggregated from the initial layers.
>
>
> Que-4: Section 4: Why was DNN-9L chosen? Probably a minor question, but unless that comes from a related work, I'm not sure where the 9 came from.
>
> Ans-4: DNN-9L is denoted as if all the predictions were made at the 9th layer of the model. This baseline is to show the reduction in performance if the layers are statically reduced instead of dynamic reduction in layers. This proves the importance of dynamic methods.
>
>
> Que-5: General: It seems intuitive that perhaps some layers are only specifically good at EE-classifying a subset of classes. Have you observed any behavior like this? It would tie into my question about equation 2, which is that because you're relying on a chain of consistent max labels, that uneven error distributions could be harming your overall approach.
>
> Ans-5: We had not observed this in text classification task as there are less than 2-5 classes. However, for the image captioning tasks, we had observed that more common tokens such as ‘the’, ‘a’ were exiting at initial layers while the rare tokens or not so common tokens such as ‘cake’, ‘knife’ were exited deeper into the backbone.
>
> We hope that we clarified your doubts, please let us know if any further clarifications are required. Once again thanks for your time and efforts in assessing our work.

---

> > ### Comment · Reviewer_mL6L · 2024-11-22
> >
> > Thank you.
> >
> > Question 3: Given that you're using the rule that accuracy must improve over the baseline, and using that to set thresholds, wouldn't that cause the $w_i$ to be small for early layers with potentially spurious representations? To be clear, I don't think this is a disqualifier for the work presented in the paper, but rather whether you've considered the approach, and if so, verified that it's worse than maxpred due to false confidence.
> >
> > Question 4: I suppose the question I was asking was "why 9 specifically". Looking at table 3 again, I suppose it's because the 5 comparison (x)BERT methods all have speedup ratios around 1.33, so their expected exit point is roughly layer 9. For table 1, I think the choice of 9 looks more arbitrary as most methods have higher speedup ratios, and so a choice of something achieving a speedup ratio of around 1.7-1.8 would seem more suitable in the ALBERT group. Similarly for table 2, a choice of speedup around 1.6x might seem more natural.
> >
> > Overall, I'm still inclined to keep my current score the same.

---

> > > ### Author Response · Authors · 2024-11-22
> > > **Further clarifications**
> > >
> > > Thanks for the acknowledgement
> > >
> > > Sure, your suggestion is logical and that could also be explored. But please note that your suggestion is a different way of combining the confidence scored of individual classifiers. Multiple such combinations can be defined. Still, we will explore the idea given by you.
> > >
> > >
> > > For the second ques: We chose 9th layer specifically as for the image captioning task the performance of the 6 th layer was really poor and the results made no sense to compare with the final layer. For 9th layer, it was pretty better and comparable.
> > >
> > > Once again thanks for the acknowledgement.

---

### Official Review · Reviewer_RQs8 · 2024-11-02

**Soundness:** 3
**Presentation:** 1
**Contribution:** 3
**Rating:** 6
**Confidence:** 4

**Summary:**

The authors propose a new decision criterion, BEEM, where exit classifiers are considered experts whose confidence scores are aggregated. A sample exits when the aggregated confidence surpasses a set threshold. The paper evaluates this approach on the COCO dataset for image captioning and GLUE datasets for various transformer-based language tasks.

**Strengths:**

The results look promising, as shown in Tables 1, 2, and 3.

As this area is outside my expertise, I would defer to my fellow reviewers on:
- The relevance of the benchmark tasks and datasets used in the paper
- The significance of the results
- Any potential biases or issues within the experimental setup

**Weaknesses:**

Presentation issues:
- The introduction and abstract delve too deeply into technical details from the outset. Consider starting with a high-level overview before moving into specifics, which are available in detail later.
- Rework Figure 1: It should be consistent with other figures. For instance, S1 should connect to S2 with an arrow.
- The title of Section 3 ("Problem Setup") feels off.
- Section 3.1: The setup is challenging to follow. How are parameters denoted for exit classifiers? Are they trained independently or jointly with the model? Are the exit classifiers linear? Additionally, a brief (one-sentence) explanation of why the KL term is relevant would - improve clarity. Is the inference algorithm novel here, or adapted from previous work?
Section 3.2: Describe the task (e.g., image captioning) first before discussing specific architectures.
- Line 219: Statements like "confidence-based early exit methods like DeeBERT and ElasticBERT..." lack supporting evidence. Many claims are subjective, without empirical or theoretical backing.
- Section 3.3: The cost vector isn't "learned" in the typical sense; it's set by the model developer. Adjust the wording accordingly.
- Lines 256–259 are confusing; rephrase them. For instance, what is L? Is it the number of layers? Why is S_i < L important? What does i signify?
- In Section 3, contributions are not clearly separated from known results in the literature. Improved subsectioning could help delineate the original contributions.

Nit:
- Use \citep instead of \citet when appropriate (e.g., Line 46).
- Line 215: There's no Equation 5.
- Theorem 3.1: Avoid using t for the layer index, as it was previously used for tokens (Line 205). Clarify terms a_t and b_t, which are hard to interpret.
- Table 5 should appear after Table 4.

**Questions:**

The paper seems somewhat rushed, with presentation issues that would benefit from an additional round of revision (resubmission basically). From my past experiences, these are unlikely to be fully addressed during the rebuttal period. Given the positive empirical results, I believe the paper would be stronger with more attention to clarity in writing, figure design, and structural organisation.

How much are you prepared to revise the presentation of the paper? If you are aiming for acceptance, please outline a specific plan to improve its readability and structure.

---

> ### Author Response · Authors · 2024-11-20
> **Rebuttal**
>
> Thanks for the comments.
>
> Ques-1: The introduction and abstract delve too deeply into technical details from the outset. Consider starting with a high-level overview before moving into specifics, which are available in detail later.
>
> Ans-1: We started off our paper with a not so basic introduction as Early Exits is one of the very prominent dynamic inference methods and started off in the 2016. There is a lot of literature and even surveys on EE. We also wanted to make our paper technically sound hence a detailed introduction was crucial.
>
> Ques-2: Rework Figure 1: It should be consistent with other figures. For instance, S1 should connect to S2 with an arrow.
>
> Ans-2: We have added an arrow in the figure 1 and is visible in the newly uploaded pdf, please have a look.
>
> Ques-3: The title of Section 3 ("Problem Setup") feels off.
>
> Ans-3: We have checked the tex file and there is no error, please have a look in the revised version if it still looks off.
>
>
>
> Ques-3 : How are parameters denoted for exit classifiers?
>
> Ans-3 : As we have stated in line 168 that $\theta$ denotes the set of all the parameters, it also consists of the parameters of the exit classifiers.
>
> Ques-4 : Are they trained independently or jointly with the model?
>
> Ans-4 : They are jointly trained with the model as seen in the objective function for training.
>
> Ques: Are the exit classifiers linear?
>
> Ans: Yes they are linear as stated in lines 392-393.
>
> Ques-5: Additionally, a brief (one-sentence) explanation of why the KL term is relevant would - improve clarity.
>
> Ans-5: We have added “KL divergence (Kullback-Leibler divergence) is used in knowledge distillation because it measures how well one probability distribution (the student model's predictions) approximates another (the teacher model's predictions). In the context of knowledge distillation, KL divergence serves as a key component to transfer "soft knowledge" from the teacher to the student.” in lines 171-174 of our revised version.
>
> Ques-6: Is the inference algorithm novel here, or adapted from previous work?
>
> Ans-6: Yes, the inference algorithm is novel as we have proposed a different method to measure the confidence metric that ensembles multiple classifiers to decide exiting. Als, we set the threshold in a different novel manner as compared to other methods.
>
> Ques-7: Describe the task (e.g., image captioning) first before discussing specific architectures.
>
> Ans-7: We have added “For the image captioning task where the objective is to generate a caption for an input image,” in lines 194 of our revised version.
>
> Ques-8: Line 219: Statements like "confidence-based early exit methods like DeeBERT and ElasticBERT..." lack supporting evidence. Many claims are subjective, without empirical or theoretical backing.
>
> Ans-8: For supporting this, we have already cited the DeeBERT and ElasticBERT paper in multiple places in the paper such as: line 150 and line 046 etc. It would be helpful if you specify the subjective claim that needs further clarifications.
>
> Ques-9: Lines 256–259 are confusing; rephrase them. For instance, what is L? Is it the number of layers? Why is S_i < L important? What does i signify?
>
>
>
> Ans-9: We have rephrased them as “ The values of $w_i\in [0,1]$, $C_i\in [0,1]$ imply that $S_i \leq L$ i.e. the score at any exit layer $i$ cannot be greater than the number of layers $L$ as $S_i$ is a multiple of two values between $0$ and $1$, it becomes very small and is added almost $L$ times. We choose the best-performing threshold on the validation set in terms of accuracy.” Please suggest if there are more changes required.
>
> Ques-10: Use \citep instead of \citet when appropriate (e.g., Line 46).
>
> Ans-10: We have updated it in the revised version submitted now.
>
> Ques-11: Line 215: There's no Equation 5.
>
> Ans-11: We fixed this in the revised version.
>
> Ques-12: Theorem 3.1: Avoid using t for the layer index, as it was previously used for tokens (Line 205). Clarify terms a_t and b_t, which are hard to interpret.
>
> Ans-12: We have fixed this in the revised version.
>
> Ques-13: Table 5 should appear after Table 4.
>
> Ans-13: Fixed this issue in the revised (newly submitted version).
>
> We have fixed most of the presentation issues, please have a look by downloading the newest version and suggest if there are any further changes required. We hope that we answered most of your doubts. Please let us know if you have any further questions. If not, we request you to please reassess the scores.

---

> > ### Comment · Reviewer_RQs8 · 2024-11-25
> >
> > LGTM, thanks for the updates. Score 5-->6

---

> ### Author Response · Authors · 2024-11-25
> **Thanks**
>
> Thanks for the acknowledgement and increasing the score.

---

### Official Review · Reviewer_9UE5 · 2024-11-04

**Soundness:** 2
**Presentation:** 2
**Contribution:** 2
**Rating:** 5
**Confidence:** 4

**Summary:**

The paper introduces BEEM, a novel framework for enhancing the performance of Early Exit Deep Neural Networks (DNNs) by treating exit classifiers as experts. The key idea is to aggregate confidence scores from these exit classifiers, making exit decisions based on the ensemble effect and consistency of predictions. A sample exits the network when the aggregated confidence exceeds a predefined threshold, which is determined by the error rates of intermediate exits. The paper presents experimental results on the COCO dataset for image captioning and GLUE datasets for language tasks, showing improvements in speed-up by a factor of 1.5× to 2.1× and comparable or improved accuracy compared to the final layer of DNNs. The significance lies in its ability to reduce inference latency while maintaining or enhancing accuracy, particularly useful for resource-constrained environments.

**Strengths:**

The paper introduces BEEM, a novel framework for enhancing the performance of Early Exit Deep Neural Networks (DNNs) by treating exit classifiers as experts. The key idea is to aggregate confidence scores from these exit classifiers, making exit decisions based on the ensemble effect and consistency of predictions. A sample exits the network when the aggregated confidence exceeds a predefined threshold, which is determined by the error rates of intermediate exits. The paper presents experimental results on the COCO dataset for image captioning and GLUE datasets for language tasks, showing improvements in speed-up by a factor of 1.5× to 2.1× and comparable or improved accuracy compared to the final layer of DNNs. The significance lies in its ability to reduce inference latency while maintaining or enhancing accuracy, particularly useful for resource-constrained environments.

**Weaknesses:**

1. The paper provides extensive experimental results demonstrating the effectiveness of BEEM in improving inference speed and accuracy across various NLP tasks and image captioning, which strengthens the credibility of the proposed method.

2. The theoretical analysis providing conditions under which BEEM outperforms standard DNN inference adds depth to the paper and offers insights into its underlying mechanisms.

3. The paper is well-organized, with clear explanations of the methodology, experiments, and results, making it accessible to readers.

**Questions:**

1. While the paper demonstrates strong performance on NLP and image captioning tasks, it is unclear how well BEEM generalizes to other types of tasks or datasets, which could be a limitation in its applicability.

2. The paper could provide more details on the computational complexity of implementing BEEM, especially regarding the aggregation of confidence scores and the potential overhead it introduces.

3. The performance of BEEM is highly dependent on the choice of thresholds, and the paper could benefit from a deeper exploration of how sensitive the model is to these choices.

---

> ### Author Response · Authors · 2024-11-20
> **Rebuttal**
>
> Thanks for the insightful comments.
>
> Q1: While the paper demonstrates strong performance on NLP and image captioning tasks, it is unclear how well BEEM generalizes to other types of tasks or datasets, which could be a limitation in its applicability.
>
> A1: We have provided results on various NLP classification tasks and image captioning tasks which are two major standard tasks in deep learning. However, note our proposed confidence metric does not depend on the model or task at hand and can be applied to any task and model. Another thing that we propose is the choice of threshold which again does not depend on the model or tasks at hand and can be used for any task and model architecture. Since there is no dependence of our method on the task type, it is easily generalizable to the tasks that the chosen model is capable of doing. To prove this, we have experimented with two different domains: NLP and image. In NLP we have considered three different tasks: entailment classification, sentiment analysis, natural language inference task.
>
> Still, if you want us to provide results on a particular type of dataset, please let us know, and we will add that into the paper.
>
>
> Q2: The paper could provide more details on the computational complexity of implementing BEEM, especially regarding the aggregation of confidence scores and the potential overhead it introduces.
>
> A2: We believe that there will be no additional computational overhead as compared to existing EE methods since, for instance consider the confidence metric by DeeBERT where the confidence is checked at every layer, we just store this confidence value, hence no additional computation. Similarly, PABEE checks the consistency in prediction, it also stores confidence values without additional computational complexity. Hence, there is no additional complexity in storing the confidence values.
>
> If you are talking about the computational cost of getting the confidence values, it is also minimal as compared to backbone layers, as it consists of only a single linear with a task to map the hidden representations to the output classes. Due to this simplicity, existing methods also do not consider the computational cost of the exits. We are using similar architectures for the exits as already used in existing methods. For fair comparison, we are using the same metrics as used by earlier methods such as speedup to decide the reduction in computational cost.
>
> Q3: The performance of BEEM is highly dependent on the choice of thresholds, and the paper could benefit from a deeper exploration of how sensitive the model is to these choices.
>
> A3: The threshold in our setup is obtained in our method by an optimization problem. The only thing that remains to explore is how sensitive is the optimization problem solution to the validation set size that we show in the table below.
>
> |              | SST-2 | | MNLI  ||
> | ------------ | ----- | ----- | ----| -----|
> | Val_set_size | Acc   | Speed | Acc | Speed |
> | 500         | 91.9  | 1.93  | 83.1 | 2.07 |
> | 1000        | 92.4  | 1.92  | 83.9 | 2.01 |
> | 2000        | 92.5  | 1.92  | 84.1 | 1.99 |
> | 4000        | 92.6  | 1.91  | 84.1 | 1.96 |
>
> Note from the table that the val set size has very small impact on the performance of the model making our method less sensitive to the val set.
> We also establish that using our threshold the overall accuracy is better than that of the last layer. Hence we provide a sound method for the choice of threshold which is missing in earlier works.
>
> An ablation study over this has been made in Table 5 and Section 6.2 in our work where we detail the effect of setting the threshold by solving the optimization problem vs if the thresholds were chosen without any objective function. If you need any other type of ablation, please let us know, and we will add it in the paper.
>
> We hope that we clarified most of your concerns and doubts. Please let us know if you have any further questions and we will be happy to answer that. However, we are surprised that you have not even raised any major concerns and given overall positive feedback to our approach, still, you gave us a rating of 5. We request that you please reassess the score if you are happy with our contribution. Again, we thank you for your time and efforts in assessing our work.

---

> ### Author Response · Authors · 2024-11-26
> **Reminder**
>
> Dear reviewer 9UE5,
>
> It is a gentle reminder to acknowledge our rebuttal. Again, we thank you for your time and effort in reviewing our work.
>
> Regards,
>
> Authors

---

> ### Author Response · Authors · 2024-11-27
> **Reminder**
>
> Dear reviewer,
>
> It is a gentle reminder to acknowledge our rebuttal. Again, we thank you for your time and effort in reviewing our work.
>
> Regards,
>
> Authors

---

> ### Author Response · Authors · 2024-11-28
> **Reminder**
>
> Dear reviewer 9UE5,
>
> It is a gentle reminder to acknowledge our rebuttal. Again, we thank you for your time and effort in reviewing our work.
>
> Regards,
>
> Authors

---

> ### Author Response · Authors · 2024-11-29
> **Reminder**
>
> Dear reviewer 9UE5,
>
> It is a gentle reminder to acknowledge our rebuttal. Again, we thank you for your time and effort in reviewing our work.
>
> Regards,
>
> Authors

---

> ### Author Response · Authors · 2024-12-01
>
> Dear reviewer 9UE5,
>
> As the discussion phase is ending soon, it is a gentle reminder to acknowledge our rebuttal.
> Again, we thank you for your time and effort in reviewing our work.
>
> Regards,
>
> Authors

---

> > ### Author Response · Authors · 2024-12-01
> > **Reminder**
> >
> > Dear reviewer 9UE5,
> >
> > As the discussion phase is ending soon, it is a gentle reminder to acknowledge our rebuttal. Again, we thank you for your time and effort in reviewing our work.
> >
> > Regards,
> >
> > Authors

---

> > > ### Author Response · Authors · 2024-12-02
> > > **Reminder**
> > >
> > > Dear reviewer 9UE5,
> > >
> > > With only 1 day remaining in the author-reviewer discussion, we request you to please acknowledge our rebuttal and let us know if you have any further concerns.
> > >
> > > Regards,
> > >
> > > Authors.

---

### Official Review · Reviewer_MFiX · 2024-11-08

**Soundness:** 3
**Presentation:** 2
**Contribution:** 2
**Rating:** 6
**Confidence:** 4

**Summary:**

This paper studies the early exit (EE) technologies for language tasks and image captioning tasks. More specifically, instead of making decisions based on individual intermediate layers, this paper proposed to use weighted sum of multiple intermediate layers to achieve stronger results. Thresholds for each layer are set with error rate restrictions on the validation set, to make sure each exit classifier (on early exited examples) performs no worse than the final layer. Speed-up improvements are achieved on both language and image captioning tasks, while on language tasks there’s also a tiny quality boost.

**Strengths:**

* Extensive experiments are conducted on both language tasks and the image captioning task, with source code attached and plenty of baseline results reported.
* The proposed early exit method not only speeds up the inference, but also improves model quality on language tasks.

**Weaknesses:**

* Speed-up factor is only computed based on the theoretical analysis (line 412), which is very misleading, especially for image captioning tasks where exit is only attached to the decoder but not encoder (line 201). In this case, do you count speed up only based on decoder layers? This is problematic, for example the baseline method MuE in Table3 applies early exit to both the encoder and the decoder (Figure2 of https://arxiv.org/pdf/2211.11152) and how is that calculated in line 386? Or do you also count encoder layers as a constant in this case as the inference cost is never reduced for encoders? If that’s the case, then it’s also not accurate due to the shape mismatch between the vision encoder and the language decoder. It would be more useful to report the actual end-to-end inference time speed up with and without early exit.
* Per-layer threshold tuning depends on the validation set of each downstream task, which makes it not practical for broad use cases (e.g. zero-shot or few-shot tasks). Section 6.2 compares vallina threshold selection with the proposed threshold selection method, do you also have study showing the impact of validation set size? For example, if reduce the size of the validation set, would that significantly affect the proposed threshold setting method quality?
* The proposed method is incremental, where it differs from the baseline methods of whether or not to use a weighted sum of the exit classifiers. And there’s also no ablation test to verify this point of using a weighted sum versus using simply individual exit classifiers. It's unclear whether we should treat DeeBERT and PABEE as ablation tests because the results are produced with the existing code (line 376) instead of a comparable setup (e.g. exactly the same code from this paper).

**Questions:**

* In line 412, to calculate speed-up factors for image captioning tasks, how do you distinguish the cost from the vision encoder and the language encoder / decoder?
* Are the vision / language backbones fine-tuned or frozen (where only the new linear layers are trainable)?

---

> ### Author Response · Authors · 2024-11-20
> **Rebuttal**
>
> Thanks for the insightful comments.
>
> Q1: Speed-up factor is only computed based on the theoretical analysis (line 412), which is very misleading, especially for image captioning tasks where exit is only attached to the decoder but not encoder (line 201). In this case, do you count speed up only based on decoder layers? This is problematic, for example the baseline method MuE in Table3 applies early exit to both the encoder and the decoder (Figure2 of https://arxiv.org/pdf/2211.11152) and how is that calculated in line 386? Or do you also count encoder layers as a constant in this case as the inference cost is never reduced for encoders? If that’s the case, then it’s also not accurate due to the shape mismatch between the vision encoder and the language decoder. It would be more useful to report the actual end-to-end inference time speed up with and without early exit.
>
> A1: **Speedp metric:** We acknowledge that the end-to-end time would have been a better metric but we have considered the Speedup factor due to the facts that 1) we wanted to be consistent with the existing methods such as DeeCAP and MuE. 2) Also, it can easily be converted into multiple metrics like expected time and also proportional to wall clock time. Existing methods such as DeeCAP, MuE, PABEE, DeeBERT etc. already explain its benefits that made it a standard metric for judging Early Exit speedup.
>
> **Why only Decoder Speedup:** The encoder speed is always a constant value i.e. 1 as there is no speedup observed in the encoder. It does not depend on the shape of the encoder and could be considered constant across all the baselines (as done by existing baselines except MuE).
>
> Note that, the encoder cost is very minimal as compared to the decoder in the image captioning tasks with autoregressive decoding as after one pass on the encoder, the decoder makes multiple forward passes until the end-of-sentence token is predicted, hence in autoregressive decoding the major part where we need to reduce the inference time is by adding exits to the decoder.
>
> Finally, the early exit approach in MuE cannot be extended to the models where the shape of the output of the encoder or decoder changes at every layer as we cannot compute the similarity score. This is the case with our setup as well. The Swin Transformer is used as an encoder in our method, which has a different shape output at every layer, hence the MuE approach could not be extended to our encoder.
>
>
>
> Q2: Per-layer threshold tuning depends on the validation set of each downstream task, which makes it not practical for broad use cases (e.g. zero-shot or few-shot tasks). Section 6.2 compares vallina threshold selection with the proposed threshold selection method, do you also have study showing the impact of validation set size? For example, if reduce the size of the validation set, would that significantly affect the proposed threshold setting method quality?
>
> A2: Validation set for fine-tuning thresholds: Yes, for the early exit models, we require a training dataset, as the weights of the classifiers need to be learned, these models require some amount of fine-tuning to train the weights of the attached early exits. To the best of our knowledge, most of the works perform fine-tuning of the backbone with exits before the inference, which requires a training set. From that training dataset, we take a small split of the dataset as validation that is used for setting the threshold.
>
> **Ablation study: **Thanks for the suggestion about the ablation study on the size of the validation dataset, here is a table that contains the required results.
>
> |              | SST-2 | | MNLI  ||
> | ------------ | ----- | ----- | ----| -----|
> | Val_set_size | Acc   | Speed | Acc | Speed |
> | 500         | 91.9  | 1.93  | 83.1 | 2.07 |
> | 1000        | 92.4  | 1.92  | 83.9 | 2.01 |
> | 2000        | 92.5  | 1.92  | 84.1 | 1.99 |
> | 4000        | 92.6  | 1.91  | 84.1 | 1.96 |
>
>
> The table shows the impact on the performance when the validation set size is reduced of the training dataset. Note from the table that our method of setting threshold is not too sensitive on the size of the validation set until it is extremely small such as only 500 samples used as the validation dataset. The drop in such cases is due to the fact that the optimization algorithm gets a very small number of samples for optimality in general. However, note that we expect that with such large datasets available we believe that the size of the validation set will be greater than 500 samples.
>
> More clarifications follow in the reply to this.

---

> ### Author Response · Authors · 2024-11-20
> **Further clarifications.**
>
> Q3: The proposed method is incremental, where it differs from the baseline methods of whether or not to use a weighted sum of the exit classifiers. And there’s also no ablation test to verify this point of using a weighted sum versus using simply individual exit classifiers. It's unclear whether we should treat DeeBERT and PABEE as ablation tests because the results are produced with the existing code (line 376) instead of a comparable setup (e.g. exactly the same code from this paper).
>
> A3: **Increment work:** We disagree that our work is incremental. Earlier work focused on proposing a confidence metric for early exit decisions without delving into threshold values. Our work considered both aspects which makes it more complete. We note that earlier works chose thresholds based on user choice or based on a threshold that maximizes the accuracy over the validation set. In our case, we have formulated this problem as an optimization problem with a constraint. The constraint ensures overall accuracy with early exits, which is better than exiting from the final layer only. Further, we have given a theoretical analysis of this claim. Earlier works do not provide such analysis.
>
> In summary, our work provides a full solution to the early exit procedure where we first proposed a new confidence metric that uses the ensemble of classifiers to make exiting more confident and reliable. We then provided a solution to the important problem of threshold selection and gave a sound method with theoretical justification.
>
> **Ablation tests:** The results of PABEE can be considered as an ablation study as we have used a similar setup, model, parameters, and loss objectives. Also, we performed an ablation study over what you suggested and found that it was very close to the PABEE method, hence did not explicitly put a table and section for this.
>
> |                  | SST-2 | |MNLI  | |
> | ---------------- | ----- | ----- | -----   | ----- |
> |                  | Acc   | Speed | Acc | Speed |
> | Confidence-based | 89.8  | 1.78  | 81.3 | 1.7 |
> | Patience-based   | 92.3  | 1.87  | 84 | 1.85 |
> | Ours             | 92.7  | 1.98  | 84.8 | 1.96 |
>
> However, for DeeBERT, as the setup was quite different (DeeBERT performs separate training and we perform joint training) there are slight changes in the result but the PABEE result is the same as our setup is similar to PABEE. We will add these results in the appendix of the final version.
>
>
> We hope that we clarified most of your doubts and concerns, if you have any further comments, doubts or suggestions, please let us know and we will be happy to clarify them. If not, we request you to please reassess the scores. We again thank you for your time and effort in assessing our work.

---

> > ### Author Response · Authors · 2024-11-26
> > **Reminder**
> >
> > Dear reviewer MFiX,
> >
> > It is a gentle reminder to acknowledge our rebuttal and make necessary reassessment of the score. Again, we thank you for your time and effort in reviewing our work.
> >
> > Regards,
> >
> > Authors

---

> > > ### Author Response · Authors · 2024-11-27
> > > **Reminder**
> > >
> > > Dear reviewer,
> > >
> > > It is a gentle reminder to acknowledge our rebuttal and make necessary reassessment of the score. Again, we thank you for your time and effort in reviewing our work.
> > >
> > > Regards,
> > >
> > > Authors

---

> ### Comment · Reviewer_MFiX · 2024-11-28
>
> Thanks for the response and supporting results of various validation set sizes. They addressed my concerns, thus I increase my score from 5 to 6.

---

> > ### Author Response · Authors · 2024-12-03
> > **Thanks**
> >
> > Thanks for acknowledgement of the rebuttal and increasing the score.

---

### Author Response · Authors · 2024-11-22
**Reminder**

Dear reviewers,

As the discussion phase is ending soon, we request you to acknowledge our rebuttal and let us know if you have any further questions.

Regards,

Authors

---

### Author Response · Authors · 2024-11-24
**Gentle reminder**

Dear reviewers,

As the discussion phase is ending soon, it is a gentle reminder to acknowledge our rebuttal and make the necessary changes based on that.

Again, we thank you for your time and effort in reviewing our work.

Regards,

Authors

---

### Meta-Review · Area_Chair_EXgp · 2024-12-21

**Metareview:**

This paper proposes BEEM (Boosting Performance of Early Exit DNNs using Multi-Exit Classifiers as Experts), a new decision criterion for "early exit" (instead of using the full forward pass, if the intermediate output is sufficiently good, then use the intermediate output as the final output). BEEM treats the exit classifiers as experts and aggregates their predictions. Based on the consistency of the predictions and threshold, this paper set the early exit of DNN. The empirical verification is conducted on COCO and GLUE, where the proposed method shows both benchmark scores and speed-up improvements.

This paper has mixed opinions (three borderlines and one positive), where the reviewer who recommended borderline rejection did not engage in the discussion despite the reminders from the authors and the AC. This paper has strengths in (1) achieving meaningful improvements in inference speed while keeping the original accuracy and (2) the novelty of the idea. There were several concerns from the reviewers. For example:

- [RQs8, mL6L] Writing quality
- [MFiX, 9UE5] No detailed study on the sensitivity of the threshold selection / validation set size /  Patience-based vs. confidence-based
- [MFiX] A non-realistic speed-up metric and decoder-only exit scenario would be problematic.
- [9UE5] The tasks would not be generalized to the other tasks
- [9UE5] The aggregation of confidence scores may need additional computational complexity.

Reviewer RQs8, mL6L and MFiX acknowledged that most of their concerns were resolved after reading the authors' responses. Specifically, the authors provided ablation studies on the validation set size and the design choice (Patience-based vs. confidence-based).

Regarding the raised concerns by Reviewer 9UE5, I think that some of them are resolved and the others are still unresolved. For example, the sensitivity of the threshold selection would be addressed by the validation set size ablation study, and the complexity issue was clarified by the authors (no additional complexity in storing the confidence values compared to previous EE methods).

On the other hand, I also agree with Reviewer 9UE5 that it would be great to validate the proposed method on different domains and tasks. For example, one can test the benefit of the proposed method on supervised vision classification (e.g., ImageNet) or zero-shot vision classification (e.g., CLIP) that does not need a decoder framework. As pointed out by Reviewer MFiX, this work only considers the decoder-only exit strategy, and as the authors acknowledged that "the encoder cost is very minimal", the proposed method would not be effective in encoder-only tasks as much as encoder-decoder frameworks. However, this would not be a case for rejection, but a gentle suggestion for improving the submission; I recommend adding more experiments on encoder-only scenarios to improve its real-world contribution.

Overall, all the reviewers who engaged in the discussion reached a positive consensus, and the opinion from the negative reviewer looks a manageable issue. I recommend acceptance for this paper.

**Additional Comments On Reviewer Discussion:**

There were several concerns from the reviewers. For example:

- [RQs8, mL6L] Writing quality
- [MFiX, 9UE5] No detailed study on the sensitivity of the threshold selection / validation set size /  Patience-based vs. confidence-based
- [MFiX] A non-realistic speed-up metric and decoder-only exit scenario would be problematic.
- [9UE5] The tasks would not be generalized to the other tasks
- [9UE5] The aggregation of confidence scores may need additional computational complexity.

Reviewer RQs8, mL6L and MFiX acknowledged that most of their concerns were resolved after reading the authors' responses. Specifically, the authors provided ablation studies on the validation set size and the design choice (Patience-based vs. confidence-based).

---

### Decision · Program_Chairs · 2025-01-22

Accept (Poster)